# PERCEIVER-ACTOR:
# A Multi-Task Transformer for Robotic Manipulation

**Mohit Shridhar** [1,*]   **Lucas Manuelli** [2]   **Dieter Fox** [1,2]
[1]University of Washington    [2]NVIDIA
mshr@cs.washington.edu   lmanuelli@nvidia.com   fox@cs.washington.edu

[peract.github.io](peract.github.io)

**Abstract:** Transformers have revolutionized vision and natural language processing with their ability to scale with large datasets. But in robotic manipulation, data is both limited and expensive. Can manipulation still benefit from Transformers with the right problem formulation? We investigate this question with PERACT, a language-conditioned behavior-cloning agent for multi-task 6-DoF manipulation. PERACT encodes language goals and RGB-D voxel observations with a Perceiver Transformer [1], and outputs discretized actions by "detecting the next best voxel action". Unlike frameworks that operate on 2D images, the voxelized 3D observation and action space provides a strong structural prior for efficiently learning 6-DoF actions. With this formulation, we train a single multi-task Transformer for 18 RLBench tasks (with 249 variations) and 7 real-world tasks (with 18 variations) from just a few demonstrations per task. Our results show that PERACT significantly outperforms unstructured image-to-action agents and 3D ConvNet baselines for a wide range of tabletop tasks.

**Keywords:** Transformers, Language Grounding, Manipulation, Behavior Cloning

## 1   Introduction

Transformers [2] have become prevalent in natural language processing and computer vision. By framing problems as sequence modeling tasks, and training on large amounts of diverse data, Transformers have achieved groundbreaking results in several domains [3, 4, 5, 6]. Even in domains that do not conventionally involve sequence modeling [7, 8], Transformers have been adopted as a *general* architecture [9]. But in robotic manipulation, data is both limited and expensive. Can we still bring the power of Transformers to 6-DoF manipulation with the right problem formulation?

Language models operate on sequences of tokens [10], and vision transformers operate on sequences of image patches [4]. While pixel transformers [11, 1] exist, they are not as data efficient as approaches that use convolutions or patches to exploit the 2D structure of images. Thus, while Transformers may be domain agnostic, they still require the right problem formulation to be data efficient. A similar efficiency issue is apparent in behavior-cloning (BC) agents that directly map 2D images to 6-DoF actions. Agents like Gato [9] and BC-Z [12, 13] have shown impressive multi-task capabilities, but they require several weeks or even months of data collection. In contrast, recent works in reinforcement-learning like C2FARM [14] construct a voxelized observation and action space to efficiently learn visual representations of 3D actions with 3D ConvNets. Similarly, in this work, we aim to exploit the 3D structure of *voxel patches* for efficient 6-DoF behavior-cloning with Transformers (analogous to how vision transformers [4] exploit the 2D structure of image patches).

To this end, we present PERACT (short for PERCEIVER-ACTOR), a language-conditioned BC agent that can learn to imitate a wide variety of 6-DoF manipulation tasks with just a few demonstrations per task. PERACT encodes a sequence of RGB-D voxel patches and predicts discretized translations, rotations, and gripper actions that are executed with a motion-planner in an observe-act loop. PERACT is essentially a classifier trained with supervised learning to *detect actions* akin to prior work like CLIPort [16, 17], except our observations and actions are represented with 3D voxels instead of 2D image pixels. Voxel grids are less prevalent than images in end-to-end BC approaches often due

---

[*]Work done partly while the author was a part-time intern at NVIDIA.

6th Conference on Robot Learning (CoRL 2022), Auckland, New Zealand.

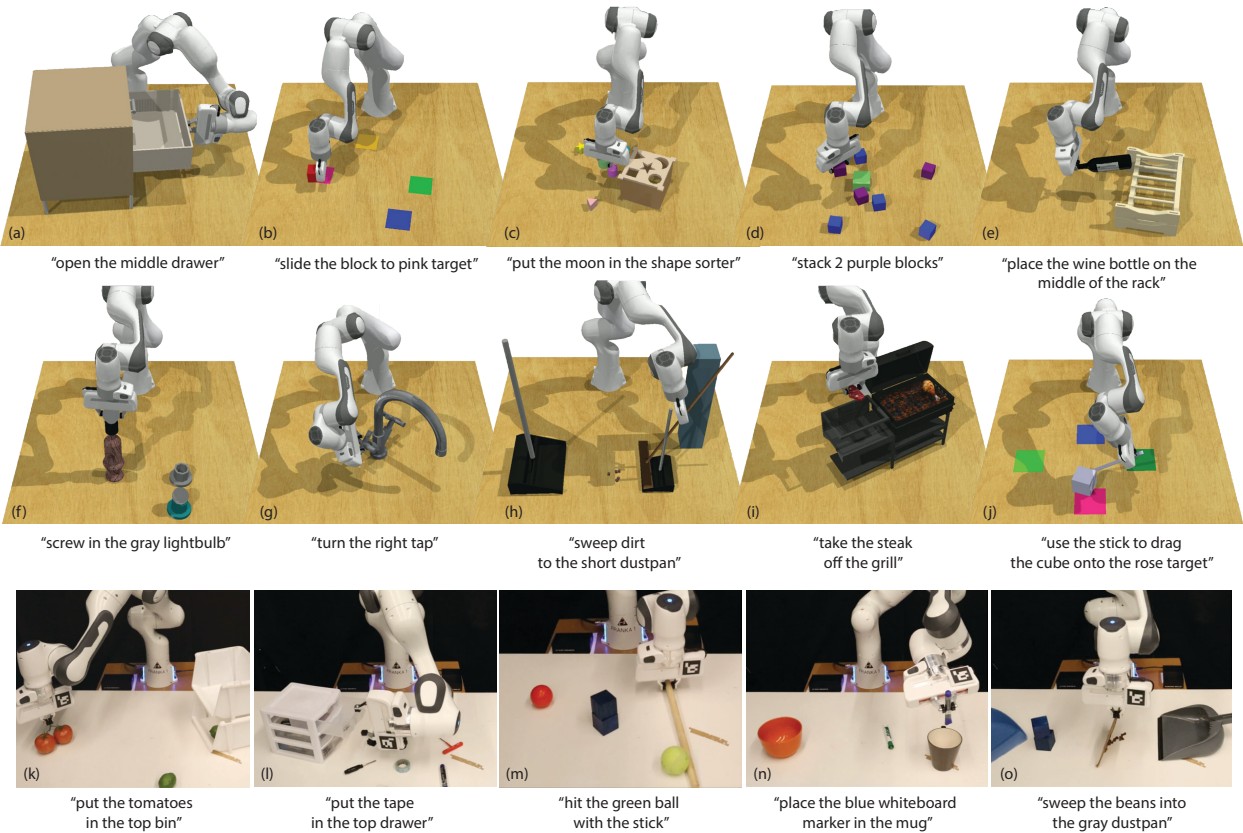

*Figure 1.* **Language-Conditioned Manipulation Tasks:** PERACT is a language-conditioned multi-task agent capable of imitating a wide range of 6-DoF manipulation tasks. We conduct experiments on 18 simulated tasks in RLBench [15] (a-j; only 10 shown), with several pose and semantic variations. We also demonstrate our approach with a Franka Panda on 7 real-world tasks (k-o; only 5 shown) with a multi-task agent trained with just 53 demonstrations. See the supplementary video for simulated and real-world rollouts.

to scaling issues with high-dimensional inputs. But in PERACT, we use a Perceiver[2] Transformer [1] to encode very high-dimensional input of up to 1 million voxels with only a small set of latent vectors. This voxel-based formulation provides a strong structural prior with several benefits: a natural method for fusing multi-view observations, learning robust action-centric[3] representations [18, 19], and enabling data augmentation in 6-DoF – all of which help learn generalizable skills by focusing on *diverse* rather than narrow multi-task data.

To study the effectiveness of this formulation, we conduct large-scale experiments in the RL-Bench [15] environment. We train a single multi-task agent on 18 diverse tasks with 249 variations that involve a range of prehensile and non-prehensile behaviors like placing wine bottles on a rack and dragging objects with a stick (see Figure 1 a-j). Each task also includes several pose and semantic variations with objects that differ in placement, color, shape, size, and category. Our results show that PERACT significantly outperforms image-to-action agents (by $34\times$) and 3D ConvNet baselines (by $2.8\times$), without using any explicit representations of instance segmentations, object poses, memory, or symbolic states. We also validate our approach on a Franka Panda with a multi-task agent trained *from scratch* on 7 real-world tasks with a **total of just 53 demonstrations** (see Figure 1 k-o).

In summary, our contributions are as follows:

- A **novel problem formulation** for perceiving, acting, and specifying goals with Transformers.
- An efficient **action-centric** framework for **grounding language in 6-DoF actions**.
- **Empirical results** investigating multi-task agents on a range of simulated and real-world tasks.

The code and pre-trained models will be made available at `peract.github.io`.

---

[2]Throughout the paper we refer to PerceiverIO [1] as Perceiver for brevity.

[3]Action-centric refers to a system that learns perceptual representations of actions; see Appendix J.

## 2 Related Work

**Vision for Manipulation.** Traditionally, methods in robot perception have used explicit "object" representations like instance segmentations, object classes, poses [20, 21, 22, 23, 24, 25]. Such methods struggle with deformable and granular items like cloths and beans that are hard to represent with geometric models or segmentations. In contrast, recent methods [26, 17, 16, 27] learn action-centric representations without any "objectness" assumptions, but they are limited to top-down 2D settings with simple pick-and-place primitives. In 3D, James et al. proposed C2FARM [14], an action-centric reinforcement learning (RL) agent with a coarse-to-fine-grain 3D-UNet backbone. The coarse-to-fine-grain scheme has a limited receptive field that cannot look at the entire scene at the finest level. In contrast, PERACT learns action-centric representations with a global-receptive field through a Transformer backbone. Also, PERACT does BC instead of RL, which enables us to easily train a multi-task agent for several tasks by conditioning it with language goals.

**End-to-End Manipulation** approaches [28, 29, 30, 31] make the least assumptions about objects and tasks, but are often formulated as an image-to-action prediction task. Training directly on RGB images for 6-DoF tasks is often inefficient, generally requiring several demonstrations or episodes just to learn basic skills like rearranging objects. In contrast, PERACT uses a voxelized observation and action space, which is dramatically more efficient and robust in 6-DoF settings. While other works in 6-DoF grasping [32, 33, 34, 35, 36, 37] have used RGB-D and pointcloud input, they have not been applied to sequential tasks or used with language-conditioning. Another line of work tackles data inefficiency by using pre-trained image representations [16, 38, 39] to bootstrap BC. Although our framework is trained from scratch, such pre-training approaches could be integrated together in future works for even greater efficiency and generalization to unseen objects.

**Transformers for Agents and Robots.** Transformers have become the prevalent architecture in several domains. Starting with NLP [2, 3, 40], recently in vision [4, 41], and even RL [8, 42, 43]. In robotics, Transformers have been applied to assistive teleop [44], legged locomotion [45], path-planning [46, 47], imitation learning [48, 49], morphology controllers [50], spatial rearrangement [51], and grasping [52]. Transformers have also achieved impressive results in multi-domain settings like in Gato [9] where a single Transformer was trained on 16 domains such as captioning, language-grounding, robotic control etc. However, Gato relies on extremely large datasets like 15K episodes for block stacking and 94K episodes for Meta-World [53] tasks. Our approach might complement agents like Gato, which could use our 3D formulation for greater efficiency and robustness.

**Language Grounding for Manipulation.** Several works have proposed methods for grounding language in robot actions [54, 55, 56, 57, 58, 59, 60, 61, 62, 63, 64, 65]. However, these methods use disentangled pipelines for perception and action, with the language primarily being used to guide perception [66]. Recently, a number of end-to-end approaches [13, 12, 67, 68, 69] have been proposed for conditioning BC agents with language instructions. These methods require thousands of human demos or autonomous episodes that are collected over several days or even months. In contrast, PERACT can learn robust multi-task policies with just a few minutes of training data. For benchmarking, several simulation environments exist [70, 17, 53], but we use RLBench [15] for its diversity of 6-DoF tasks and ease of generating demonstrations with templated language goals.

## 3 PERCEIVER-ACTOR

PERACT is a language-conditioned behavior-cloning agent for 6-DoF manipulation. The key idea is to learn perceptual representations of actions conditioned on language goals. Given a voxelized reconstruction of a scene, we use a Perceiver Transformer [1] to learn per-voxel features. Despite the extremely large input space $(100^3)$, Perceiver uses a small set of latent vectors to encode the input. The per-voxel features are then used to predict the next best action in terms of discretized translation, rotation, and gripper state at each timestep. PERACT relies purely on the current observation to determine what to do next in sequential tasks. See Figure 2 for an overview.

Section 3.1 and Section 3.2 describe our dataset setup. Section 3.3 describes our problem formulation with PERACT, and Section 3.4 provides details on training PERACT. Further implementation details are presented in Appendix B.

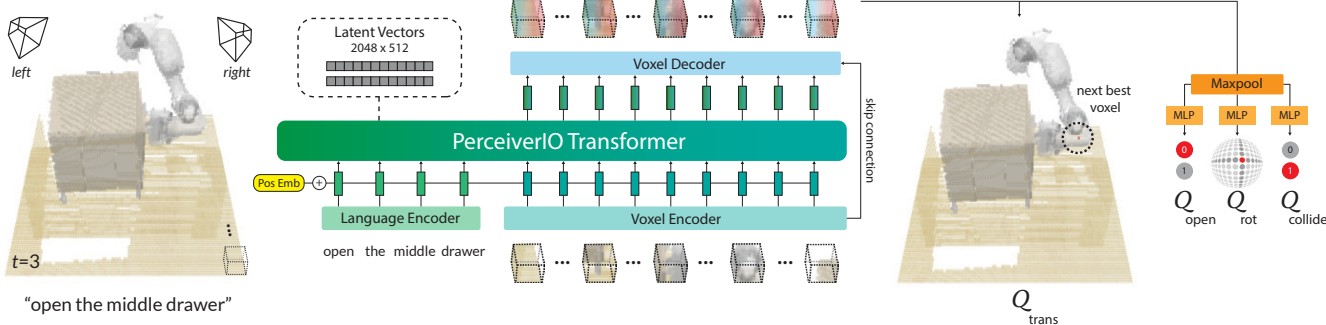

*Figure 2.* **PERACT Overview.** PERACT is a language-conditioned behavior-cloning agent trained with supervised learning to *detect actions*. PERACT takes as input a language goal and a voxel grid reconstructed from RGB-D sensors. The voxels are split into 3D patches, and the language goal is encoded with a pre-trained language model. These language and voxel features are appended together as a sequence and encoded with a Perceiver transformer [1]. Despite the extremely long input sequence, Perceiver uses a small set of latent vectors to encode the input (see Appendix Figure 6 for an illustration). These encodings are upsampled back to the original voxel dimensions with a decoder and reshaped with linear layers to predict a discretized translation, rotation, gripper open, and collision avoidance action. This action is executed with a motion-planner after which the new observation is used to predict the next discrete action in an observe-act loop until termination.

### 3.1 Demonstrations

We assume access to a dataset $\mathcal{D} = \{\zeta_1, \zeta_2, \ldots, \zeta_n\}$ of $n$ expert demonstrations, each paired with English language goals $\mathcal{G} = \{\mathbf{l}_1, \mathbf{l}_2, \ldots, \mathbf{l}_n\}$. These demonstrations are collected by an expert with the aid of a motion-planner to reach intermediate poses. Each demonstration $\zeta$ is a sequence of continuous actions $\mathcal{A} = \{a_1, a_2, \ldots, a_t\}$ paired with observations $\mathcal{O} = \{\tilde{o}_1, \tilde{o}_2, \ldots \tilde{o}_t\}$. An action $a$ consists of the 6-DoF pose, gripper open state, and whether the motion-planner used collision avoidance to reach an intermediate pose: $a = \{a_{\text{pose}}, a_{\text{open}}, a_{\text{collide}}\}$. An observation $\tilde{o}$ consists of RGB-D images from any number of cameras. We use four cameras for simulated experiments $\tilde{o}_{\text{sim}} = \{o_{\text{front}}, o_{\text{left}}, o_{\text{right}}, o_{\text{wrist}}\}$, but just a single camera for real-world experiments $\tilde{o}_{\text{real}} = \{o_{\text{front}}\}$.

### 3.2 Keyframes and Voxelization

Following prior work by James et al. [14], we construct a structured observation and action space through keyframe extraction and voxelization.

Training our agent to directly predict continuous actions is inefficient and noisy. So instead, for each demonstration $\zeta$, we extract a set of keyframe actions $\{\mathbf{k}_1, \mathbf{k}_2, \ldots, \mathbf{k}_m\} \subset \mathcal{A}$ that capture bottleneck end-effector poses [71] in the action sequence with a simple heuristic: an action is a keyframe if (1) the joint-velocities are near zero and (2) the gripper open state has not changed. Each datapoint in the demonstration $\zeta$ can then be cast as a "predict the next (best) keyframe action" task [14, 72, 73]. See Appendix Figure F for an illustration of this process.

To learn action-centric representations [18] in 3D, we use a voxel grid [74, 75] to represent both the observation and action space. The observation voxels $\mathbf{v}$ are reconstructed from RGB-D observations $\tilde{o}$ fused through triangulation $\tilde{o} \Rightarrow \mathbf{v}$ from known camera extrinsics and intrinsics. By default, we use a voxel grid of $100^3$, which corresponds to a volume of $1.0\text{m}^3$ in metric scale. The keyframe actions $\mathbf{k}$ are discretized such that training our BC agent can be formulated as a "next best action" classification task [14]. Translation is simply the closest voxel to the center of the gripper fingers. Rotation is discretized into 5 degree bins for each of the three rotation axes. Gripper open state is a binary value. Collide is also a binary value that indicates if the motion-planner should avoid everything in the voxel grid or nothing at all; switching between these two modes of collision avoidance is crucial as tasks often involve both contact based (e.g., pulling the drawer open) and non-contact based motions (e.g., reaching the handle without colliding into anything).

### 3.3 PERACT Agent

PERACT is a Transformer-based [2] agent that takes in a voxel observation and language goal $(\mathbf{v}, \mathbf{l})$, and outputs a discretized translation, rotation, and gripper open action. This action is executed with a motion-planner, after which this process is repeated until the goal is reached.

The language goal $\mathbf{l}$ is encoded with a pre-trained language model. We use CLIP's [76] language encoder, but any pre-trained language model would suffice [13, 69]. Our choice of CLIP opens up possibilities for future work to use pre-trained vision features that are aligned with the language for better generalization to unseen semantic categories and instances [16].

The voxel observation $\mathbf{v}$ is split into 3D patches of size $5^3$ (akin to vision-transformers like ViT [4]). In implementation, these patches are extracted with a 3D convolution layer with a kernel-size and stride of 5, and then flattened into a sequence of voxel encodings. The language encodings are fine-tuned with a linear layer and then appended with the voxel encodings to form the input sequence. We also add learned positional embeddings to the sequence to incorporate voxel and token positions.

The input sequence of language and voxel encodings is extremely long. A standard Transformer with $\mathcal{O}(n^2)$ self-attention connections and an input of $(100/5)^3 = 8000$ patches is hard to fit on the memory of a commodity GPU. Instead, we use the Perceiver [1] Transformer. Perceiver is a latent-space Transformer, where instead of attending to the entire input, it first computes cross-attention between the input and a much smaller set of latent vectors (which are randomly initialized and trained). These latents are encoded with self-attention layers, and for the final output, the latents are again cross-attended with the input to match the input-size. See Appendix Figure 6 for an illustration. By default, we use 2048 latents of dimension 512 : $\mathbb{R}^{2048 \times 512}$, but in Appendix G we experiment with different latent sizes.

The Perceiver Transformer uses 6 self-attention layers to encode the latents and outputs a sequence of patch encodings from the output cross-attention layer. These patch encodings are upsampled with a 3D convolution layer and tri-linear upsampling to decode 64-dimensional voxel features. The decoder includes a skip-connection from the encoder (like in UNets [77]). The per-voxel features are then used to predict discretized actions [14]. For translation, the voxel features are reshaped into the original voxel grid $(100^3)$ to form a 3D $\mathcal{Q}$-function of action-values. For rotation, gripper open, and collide, the features are max-pooled and then decoded with linear layers to form their respective $\mathcal{Q}$-function. The best action $\mathcal{T}$ is chosen by simply maximizing the $\mathcal{Q}$-functions:

$$\mathcal{T}_{\text{trans}} = \underset{(x,y,z)}{\operatorname{argmax}} \, \mathcal{Q}_{\text{trans}}((x,y,z) \mid \mathbf{v}, \mathbf{l}), \qquad \mathcal{T}_{\text{rot}} = \underset{(\psi,\theta,\phi)}{\operatorname{argmax}} \, \mathcal{Q}_{\text{rot}}((\psi,\theta,\phi) \mid \mathbf{v}, \mathbf{l}),$$

$$\mathcal{T}_{\text{open}} = \underset{\omega}{\operatorname{argmax}} \, \mathcal{Q}_{\text{open}}(\omega \mid \mathbf{v}, \mathbf{l}), \qquad \mathcal{T}_{\text{collide}} = \underset{\kappa}{\operatorname{argmax}} \, \mathcal{Q}_{\text{collide}}(\kappa \mid \mathbf{v}, \mathbf{l}),$$

where $(x,y,z)$ is the voxel location in the grid, $(\psi, \theta, \phi)$ are discrete rotations in Euler angles, $\omega$ is the gripper open state and $\kappa$ is the collide variable. See Figure 5 for examples of $\mathcal{Q}$-predictions.

### 3.4 Training Details

PERACT is trained through supervised learning with discrete-time input-action tuples from a dataset of demonstrations. These tuples are composed of voxel observations, language goals, and keyframe actions $\{(\mathbf{v}_1, \mathbf{l}_1, \mathbf{k}_1), (\mathbf{v}_2, \mathbf{l}_2, \mathbf{k}_2), \ldots\}$. During training, we randomly sample a tuple and supervise the agent to predict the keyframe action $\mathbf{k}$ given the observation and goal $(\mathbf{v}, \mathbf{l})$. For translation, the ground-truth action is represented as a one-hot voxel encoding $Y_{\text{trans}} : \mathbb{R}^{H \times W \times D}$. Rotations are also represented with a one-hot encoding per rotation axis with $R$ rotation bins $Y_{\text{rot}} : \mathbb{R}^{(360/R) \times 3}$ ($R = 5$ degrees for all experiments). Similarly, open and collide variables are binary one-hot vectors $Y_{\text{open}} : \mathbb{R}^2$, $Y_{\text{collide}} : \mathbb{R}^2$. The agent is trained with cross-entropy loss like a classifier:

$$\mathcal{L}_{\text{total}} = -\mathbb{E}_{Y_{\text{trans}}}[\log \mathcal{V}_{\text{trans}}] - \mathbb{E}_{Y_{\text{rot}}}[\log \mathcal{V}_{\text{rot}}] - \mathbb{E}_{Y_{\text{open}}}[\log \mathcal{V}_{\text{open}}] - \mathbb{E}_{Y_{\text{collide}}}[\log \mathcal{V}_{\text{collide}}],$$

where $\mathcal{V}_{\text{trans}} = \operatorname{softmax}(\mathcal{Q}_{\text{trans}}((x,y,z) \mid \mathbf{v}, \mathbf{l}))$, $\mathcal{V}_{\text{rot}} = \operatorname{softmax}(\mathcal{Q}_{\text{rot}}((\psi,\theta,\phi) \mid \mathbf{v}, \mathbf{l}))$, $\mathcal{V}_{\text{open}} = \operatorname{softmax}(\mathcal{Q}_{\text{open}}(\omega \mid \mathbf{v}, \mathbf{l}))$, $\mathcal{V}_{\text{collide}} = \operatorname{softmax}(\mathcal{Q}_{\text{collide}}(\kappa \mid \mathbf{v}, \mathbf{l}))$ respectively. For robustness, we also augment $\mathbf{v}$ and $\mathbf{k}$ with translation and rotation perturbations. See Appendix E for more details.

By default, we use a voxel grid size of $100^3$. We conducted validation tests by replaying expert demonstrations with discretized actions to ensure that $100^3$ is a sufficient resolution for execution. The agent was trained with a batch-size of 16 on 8 NVIDIA V100 GPUs for 16 days (600K iterations). We use the LAMB [78] optimizer following Perceiver [1].

For multi-task training, we simply sample input-action tuples from all tasks in the dataset. To ensure that tasks with longer horizons are not over-represented during sampling, each batch contains a uniform distribution of tasks. That is, we first uniformly sample a set of tasks of batch-size length, then pick a random input-action tuple for each of the sampled tasks. With this strategy, longer-horizon tasks need more training steps for full coverage of input-action pairs, but all tasks are given equal weighting during gradient updates.

# 4 Results

We perform experiments to answer the following questions: (1) How effective is PERACT compared to unstructured image-to-action frameworks and standard architectures like 3D ConvNets? And what are the factors that affect PERACT's performance? (2) Is the global receptive field of Transformers actually beneficial over methods with local receptive fields? (3) Can PERACT be trained on real-world tasks with noisy data?

## 4.1 Simulation Setup

We conduct our primary experiments in simulation for the sake of reproducibility and benchmarking.

**Environment.** The simulation is set in CoppeliaSim [79] and interfaced through PyRep [80]. All experiments use a Franka Panda robot with a parallel gripper. The input observations are captured from four RGB-D cameras positioned at the front, left shoulder, right shoulder, and on the wrist, as shown in Appendix Figure 7. All cameras are noiseless and have a resolution of $128 \times 128$.

**Language-Conditioned Tasks.** We train and evaluate on 18 RLBench [15] tasks. See peract.github.io for examples and Appendix A for details on individual tasks. Each task includes several variations, ranging from 2-60 possibilities, e.g., in the stack blocks task, *"stack 2 red blocks"* and *"stack 4 purple blocks"* are two variants. These variants are randomly sampled during data generation, but kept consistent during evaluations for one-to-one comparisons. Some RLBench tasks were modified to include additional variations to stress-test multi-task and language-grounding capabilities. There are a total of 249 variations across 18 tasks, and the number of extracted keyframes range from 2-17. All keyframes from an episode have the same language goal, which is constructed from templates (but human-annotated for real-world tasks). Note that in all experiments, we do not test for generalization to unseen objects, i.e., our train and test objects are the same. However during test time, the agent has to handle novel object poses, randomly sampled goals, and randomly sampled scenes with different semantic instantiations of object colors, shapes, sizes, and categories. The focus here is to evaluate the performance of a single multi-task agent trained on all tasks and variants.

**Evaluation Metric.** Each multi-task agent is evaluated independently on all 18 tasks. Evaluations are scored either 0 for failures or 100 for complete successes. There are no partial credits. We report average success rates on 25 evaluation episodes per task ($25 \times 18 = 450$ total episodes) for agents trained with $n = 10, 100$ demonstrations per task. During evaluation, an agent keeps taking actions until an oracle indicates task-completion or reaches a maximum of 25 steps.

## 4.2 Simulation Results

Table 1 reports success rates of multi-task agents trained on all 18 tasks. We could not investigate single-task agents due to resource constraints of training 18 individual agents.

**Baseline Methods.** We study the effectiveness of our problem formulation by benchmarking against two language-conditioned baselines: Image-BC and C2FARM-BC. Image-BC is an image-to-action agent similar to BC-Z [12]. Following BC-Z, we use FiLM [81] for conditioning with CLIP [76] language features, but the vision encoders take in RGB-D images instead of just RGB. We also study both CNN and ViT vision encoders. C2FARM-BC is a 3D fully-convolutional network by James et al. [14] that has achieved state-of-the-art results on RLBench tasks. Similar to our agent, C2FARM-BC also detects actions in a voxelized space, however it uses a coarse-to-fine-grain scheme to detect actions at two-levels of voxelization: $32^3$ voxels with a $1^3$m grid, and $32^3$ voxels with a $0.15^3$m grid after "zooming in" from the first level. Note that at the finest level, C2FARM-BC has a higher resolution (0.47cm) than PERACT (1cm). We use the same 3D ConvNet architecture as James et al. [14], but instead of training it with RL, we do BC with cross-entropy loss (from Section 3.4). We also condition it with CLIP [76] language features at the bottleneck like in LingUNets [82, 16].

**Multi-Task Performance.** Table 1 compares the performance of Image-BC and C2FARM-BC against PERACT. With insufficient demonstrations, Image-BC has near zero performance on most tasks. Image-BC is disadvantaged with single-view observations and has to learn hand-eye coordination from scratch. In contrast, PERACT's voxel-based formulation naturally allows for integrating multi-view observations, learning 6-DoF action representations, and data-augmentation in 3D, all of which are non-trivial to achieve in image-based methods. C2FARM-BC is the most competitive baseline, but it has a limited receptive field mostly because of the coarse-to-fine-grain scheme and partly due to the convolution-only architecture. PERACT outperforms C2FARM-BC in

| Method | open drawer | | slide block | | sweep to dustpan | | meat off grill | | turn tap | | put in drawer | | close jar | | drag stick | | stack blocks | |
|---|---|---|---|---|---|---|---|---|---|---|---|---|---|---|---|---|---|---|
| | 10 | 100 | 10 | 100 | 10 | 100 | 10 | 100 | 10 | 100 | 10 | 100 | 10 | 100 | 10 | 100 | 10 | 100 |
| Image-BC (CNN) | 4 | 4 | 4 | 0 | 0 | 0 | 0 | 0 | 20 | 8 | 0 | 8 | 0 | 0 | 0 | 0 | 0 | 0 |
| Image-BC (ViT) | 16 | 0 | 8 | 0 | 8 | 0 | 0 | 0 | 24 | 16 | 0 | 0 | 0 | 0 | 0 | 0 | 0 | 0 |
| C2FARM-BC | 28 | 20 | 12 | 16 | 4 | 0 | 40 | 20 | 60 | 68 | 12 | 4 | 28 | 24 | **72** | 24 | 4 | 0 |
| PERACT (w/o Lang) | 20 | 28 | 8 | 12 | 20 | 16 | 40 | 48 | 36 | 60 | 16 | 16 | 16 | 12 | 48 | 60 | 0 | 0 |
| PERACT | **68** | **80** | **32** | **72** | **72** | **56** | **68** | **84** | **72** | **80** | 16 | **68** | **32** | **60** | 36 | **68** | **12** | **36** |

| Method | screw bulb | | put in safe | | place wine | | put in cupboard | | sort shape | | push buttons | | insert peg | | stack cups | | place cups | |
|---|---|---|---|---|---|---|---|---|---|---|---|---|---|---|---|---|---|---|
| | 10 | 100 | 10 | 100 | 10 | 100 | 10 | 100 | 10 | 100 | 10 | 100 | 10 | 100 | 10 | 100 | 10 | 100 |
| Image-BC (CNN) | 0 | 0 | 0 | 4 | 0 | 0 | 0 | 0 | 0 | 0 | 4 | 0 | 0 | 0 | 0 | 0 | 0 | 0 |
| Image-BC (ViT) | 0 | 0 | 0 | 0 | 4 | 0 | 4 | 0 | 0 | 0 | 16 | 0 | 0 | 0 | 0 | 0 | 0 | 0 |
| C2FARM-BC | 12 | 8 | 0 | 12 | **36** | 8 | **4** | 0 | 8 | 8 | **88** | **72** | 0 | **4** | 0 | 0 | 0 | 0 |
| PERACT (w/o Lang) | 0 | **24** | 8 | 20 | 8 | **20** | 0 | 0 | 0 | 0 | 60 | 68 | **4** | 0 | 0 | 0 | 0 | 0 |
| PERACT | **28** | **24** | **16** | **44** | 20 | 12 | 0 | **16** | **16** | **20** | 56 | 48 | **4** | 0 | 0 | 0 | 0 | 0 |

*Table 1.* **Multi-Task Test Results.** Success rates (mean %) of various multi-task agents tasks trained with either 10 or 100 demonstrations per task and evaluated on 25 episodes per task. Each evaluation episode is scored either a 0 for failure or 100 for succces. PERACT outperforms C2FARM-BC [14], the most competitive baseline, with an average improvement of $1.33\times$ with 10 demos and $2.83\times$ with 100 demos.

$25/36$ evaluations in Table 1 with **an average improvement of $1.33\times$ with 10 demonstrations and $2.83\times$ with 100 demonstrations**. For a number of tasks, C2FARM-BC actually performs worse with more demonstrations, likely due to insufficient capacity. Since additional training demonstrations include additional task variants to optimize for, they might end up hurting performance.

In general, 10 demonstrations are sufficient for PERACT to achieve $> 65\%$ success on tasks with limited variations like `open drawer` (3 variations). But tasks with more variations like `stack blocks` (60 variations) need substantially more data, sometimes to simply cover all possible concepts like "*teal color block*" that might have not appeared in the training data. See the simulation rollouts in the supplementary video to get a sense of the complexity of these evaluations. For three tasks: `insert peg`, `stack cups`, and `place cups`, all agents achieve near zero success. These are very high-precision tasks where being off by a few centimeters or degrees could lead to unrecoverable failures. But in Appendix H we find that training single-task agents, specifically for these tasks, slightly alleviates this issue.

**Ablations.** Table 1 reports PERACT w/o Lang, an agent without any language conditioning. Without a language goal, the agent does not know the underlying task and performs at chance. We also report additional ablation results on the `open drawer` task in Figure 3. To summarize these results: (1) the skip connection helps train the agent slightly faster, (2) the Perceiver Transformer is crucial for achieving good performance with the global receptive field, and (3) extracting good keyframes actions is essential for supervised training as randomly chosen or fixed-interval keyframes lead to zero-performance.

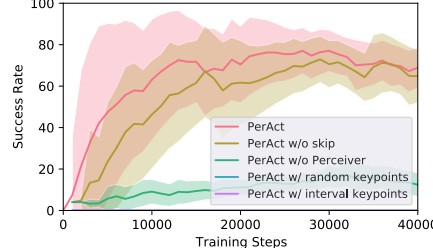

*Figure 3.* **Ablation Experiments.** Success rate of PERACT after ablating key components.

**Sensitivity Analysis.** In Appendix G we investigate factors that affect PERACT's performance: the number of Perceiver latents, voxelization resolution, and data augmentation. We find that more latent vectors generally improve the capacity of the agent to model more tasks, but for simple short-horizon tasks, fewer latents are sufficient. Similarly, with different voxelization resolutions, some tasks are solvable with coarse voxel grids like $32^3$, but some high-precision tasks require the full $100^3$ grid. Finally, rotation perturbations in the data augmentation generally help in improving robustness essentially by exposing the agent to more rotation variations of objects.

## 4.3 Global vs. Local Receptive Fields

To further investigate our Transformer agent's global receptive field, we conduct additional experiments on the `open drawer` task. The `open drawer` task has three variants: *"open the top drawer"*, *"open the middle drawer"*, and *"open the bottom drawer"*, and with a limited receptive field it is hard to distinguish the drawer handles, which are all visually identical. Figure 4 reports PERACT and C2FARM-BC agents trained with 100 demonstrations. Although the `open drawer` tasks can be

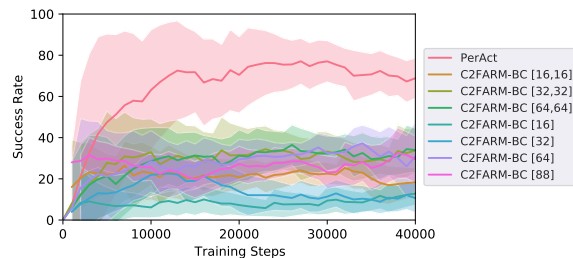

*Figure 4.* **Global vs. Local Receptive Field Experiments.** Success rates of PERACT against various C2FARM-BC [14] baselines

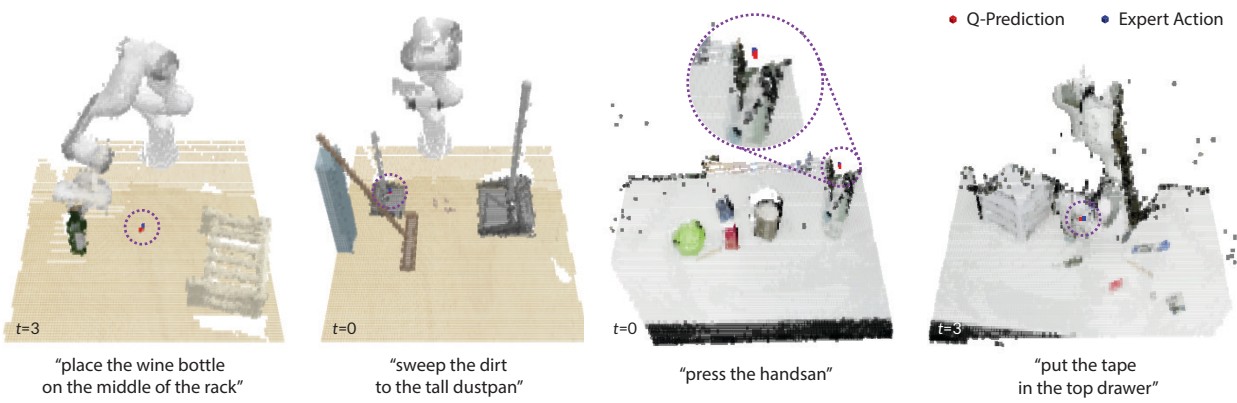

● Q-Prediction  ● Expert Action

"place the wine bottle on the middle of the rack"   "sweep the dirt to the tall dustpan"   "press the handsan"   "put the tape in the top drawer"

*Figure 5.* **Q-Prediction Examples**: Qualitative examples of translation $\mathcal{Q}$-Predictions from PERACT along with expert actions, highlighted with dotted-circles. The left two are simulated tasks, and the right two are real-world tasks. See Appendix J for more examples.

solved with fewer demonstrations, here we want to ensure that insufficient data is not an issue. We include several versions of C2FARM-BC with different voxelization schemes. For instance, [16, 16] indicates two levels of $16^3$ voxel grids at $1m^3$ and $0.15m^3$, respectively. And [64] indicates a single level of a $64^3$ voxel grid without the coarse-to-fine-grain scheme. PERACT is the only agent that achieves $> 70\%$ success, whereas all C2FARM-BC versions perform at chance with $\sim 33\%$, indicating that the global receptive field of the Transformer is crucial for solving the task.

### 4.4 Real-Robot Results

We also validated our results with real-robot experiments on a Franka Emika Panda. See Appendix D for setup details. Without any sim-to-real transfer or pre-training, we trained a multi-task PERACT agent *from scratch* on 7 tasks (with 18 unique variations) from a total of just 53 demonstrations. See the supplementary video for qualitative results that showcase the diversity of tasks and robustness to scene changes. Table 2 reports success rates from small-scale evaluations. Similar to the simulation results, we find that PERACT is able to achieve $> 65\%$ success on simple short-horizon tasks like pressing hand-sanitizers from just a handful number of

| Task | # Train | # Test | Succ. % |
|---|---|---|---|
| Press Handsan | 5 | 10 | 90 |
| Put Marker | 8 | 10 | 70 |
| Place Food | 8 | 10 | 60 |
| Put in Drawer | 8 | 10 | 40 |
| Hit Ball | 8 | 10 | 60 |
| Stack Blocks | 10 | 10 | 40 |
| Sweep Beans | 8 | 5 | 20 |

*Table 2.* Success rates (mean %) of a multi-task model trained an evaluated 7 real-world tasks (see Figure 1).

demonstrations. The most common failures involved predicting incorrect gripper open actions, which often lead the agent into unseen states. This could be addressed in future works by using HG-DAgger style approaches to correct the agent [12]. Other issues included the agent exploiting biases in the dataset like in prior work [16]. This could be addressed by scaling up expert data with more diverse tasks and task variants.

## 5 Limitations and Conclusion

We presented PERACT, a Transformer-based multi-task agent for 6-DoF manipulation. Our experiments with both simulated and real-world tasks indicate that the right problem formulation, i.e., detecting voxel actions, makes a substantial difference in terms of data efficiency and robustness.

While PERACT is quite capable, extending it to dexterous continuous control remains a challenge. PERACT is at the mercy of a sampling-based motion-planner to execute discretized actions, and is not easily extendable to N-DoF actuators like multi-fingered hands. See Appendix L for an extended discussion on PERACT's limitations. But overall, we are excited about scaling up robot learning with Transformers by focusing on *diverse* rather than narrow multi-task data for robotic manipulation.

### Acknowledgments

We thank Selest Nashef and Karthik Desingh for their help with the Franka setup at UW. We thank Stephen James for helping with RLBench and ARM issues. We are also grateful to Valts Blukis, Zoey Chen, Markus Grotz, Aaron Walsman, and Kevin Zakka, for providing feedback on the initial draft. And thanks to Shikhar Bahl for initial discussions. This work was funded in part by ONR under award #1140209-405780. Mohit Shridhar is supported by the NVIDIA Graduate Fellowship, and was also a part-time intern at NVIDIA throughout the duration of this project.

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
