# OpenReview forum: "Perceiver-Actor: A Multi-Task Transformer for Robotic Manipulation"
_robot-learning.org/CoRL/2022/Conference — CoRL 2022 Poster_

### Official Review · Reviewer_YHWS · 2022-07-27

**Originality:** Good
**Technical Quality:** Very Good
**Clarity Of Presentation:** Very Good
**Impact:** 4

**Recommendation:**

Weak Accept: I recommend accepting the paper, but will not argue for my recommendation if the majority of other reviewers have a different opinion.

**Summary:**

This paper presents a transformer-based language-conditioned multi-task imitation learning method for 6-DoF manipulation. Unlike prior language-conditioned multi-task BC methods operated on 2D images, this paper encodes both the language instructions and the RGB-D voxel observations. The authors show that such a method improves over prior approaches in both simulated and real-world robotic manipulation tasks.

**Issues:**

1. Compare to SayCan.

2. Add baseline results in the real-world experiments.

**Quality Of The Limitations Section:**

Limitations are addressed clearly

**Reviewer Expertise:**

4: The reviewer is confident but not absolutely certain that the evaluation is correct

**Robotics Focus:**

Sufficient demonstration on hardware

**Strengths And Weaknesses:**

**Strengths:**

1. This paper presents a new language-conditioned multi-task BC method operated on RGB-D voxels. The authors utilize the Perceiver to handle the high-dimensional observation space, which seems novel.

2. The empirical results appear rather convincing. The method is able to outperform prior methods such as Image-BC (similar to BC-Z) and C2FARM-BC in the multi-task simulated robotic manipulation tasks with limited demonstrations. The authors also show that the method can work well on the real robot.

3. The paper is well-written and easy to understand.

**Weakness:**

1. The paper extends prior works such as SayCan and BC-Z to the setting with the extra RGB-D voxel in the observations but the main technique is not too different from prior works, which makes the contribution a bit less novel.

2. The paper does not compare to SayCan in the experiments, which seems to be necessary. Moreover, there are no baseline results in the real-world setting.

**Summary Of Recommendation:**

Based on the summary in the Strengths And Weaknesses section, I think while the paper lacks some additional comparisons and is a bit incremental, the good empirical results and novel usage of the Perceiver architecture make senses and I would give the paper a weak accept.

---

> ### Author Response · Authors · 2022-08-22
> **Response to Reviewer-YHWS**
>
> We thank the reviewer for their feedback. We are glad the reviewer finds the empirical results very convincing and that the paper is easy to understand.
>
> **“Extends SayCan and BC-Z … and not too different from prior work”**
>
> We are not sure what is meant here by “extends SayCan and BC-Z”. To clarify, SayCan combines pre-trained skills from BC-Z with a language model that reasons about feasible high-level plans for sequential tasks. The language-based planning component of SayCan is orthogonal to both BC-Z and PerAct. Future works could use SayCan’s language model (PaLM) with PerAct’s language-conditioned policies for executing a sequence of PerAct’s skills. But this is entirely out-of-scope for this paper, which is not concerned with language-based planning. With regards to BC-Z, the Image-BC baseline reported in the paper (see L227) is the closest equivalent to BC-Z’s architecture. Our results show that PerAct significantly outperforms BC-Z, which mostly achieves zero-performance with limited demonstrations. We acknowledge that the paper could have done a better job of highlighting that Image-BC is closely related to BC-Z. We will update the paper to reflect this.
>
> **No comparisons against SayCan**
>
> Firstly, as stated above, SayCan’s language-based planning is orthogonal to PerAct. PerAct’s key premise is “getting Transformers to work for BC in the real world with limited data and high-dimensional input”, which does not involve any language-based planning. Secondly, SayCan uses BC-Z’s manipulation architecture, which we investigated with Image-BC in Table 2. Thirdly, SayCan’s and BC-Z’s complete infrastructure is not reproducible. We do not have access to SayCan’s robot dataset, robot hardware, scenes, objects, Google engineers, financial resources, or the time to collect 68000 demonstrations over the span of several months.
>
> We acknowledge that citing SayCan in the introduction might have led to some confusion. We will update the introduction to be more clear.
>
> **No comparisons in real-robot experiments**
>
> Please note that doing one-to-one comparisons in real-world experiments is difficult and often infeasible. Getting the exact same initial conditions across evaluations with objects, object poses, lighting conditions etc. is nearly impossible. For instance, in “put the tomato in the top bin”, every time the manipulator grabs the tomato wine and drops it in the bin, the tomato and the wine get deformed. Additionally, conducting one-to-one comparisons across a wide range of tasks and variations can be quite tedious and time consuming. **We have demonstrated strong results in simulation**, **which are fair and reproducible one-to-one comparisons**. Further, the physical fidelity of the simulator is sufficient to ensure that artifacts in the simulation itself aren’t being exploited to gain performance (which is a common issue in RL evaluations).
>
> **Limited novelty**
>
> In PerAct, we train a Transfromer _from scratch_ on several tasks with an input of 1 million voxels. And it works in the real-world! We are not aware of any prior work that got Transformers to work robustly for manipulation with such limited data, on such high-dimensional inputs, and on such diverse tasks. Additionally, in RLBench, PerAct outperforms a SOTA method (C2FARM) on 18 tasks (249 variations) **by 1.68x (+168%) with 10 demos and 2.66x (+266%) with 100 demos **as stated in L244**.** We hope 2.66x improvement over prior state-of-the-art across 18 tasks counts for something more than “just incremental”. We acknowledge that the paper could have done a better job of highlighting these capabilities and performance gains. And we will update the paper to fix this.

---

> > ### Comment · Reviewer_YHWS · 2022-08-28
> > **Response to the authors**
> >
> > Thank you for the response! I think most of my concerns are addressed, though I believe that real-world experiments are important to include. While I agree that it is hard to conduct head-to-head comparisons on the real robot, it is pivotal to show some success signals on the real hardware and compare the method to prior methods in a reasonable set-up (without requiring to reproduce the settings in prior works).

---

> > > ### Author Response · Authors · 2022-08-28
> > > **Thanks and Quick Question**
> > >
> > > Thank you for the comment! We are glad most of the concerns were addressed.
> > >
> > > _One quick question_: In "compare the method to prior methods" are you referring to BC-Z or SayCan here? To clarify, BC-Z and SayCan are image-to-action agents that take RGB input and output continuous 6-DoF actions. Our agent is trained with just a few minutes of real-world data – like with **10mins of data** for this ["press the handsan"](https://peract.github.io/media/intro/1_handsan.mp4) task. It's practically impossible to get image-to-action agents like BC-Z to work with 10mins of training data. We could potentially run this baseline next week, but we don't expect anything other than zero performance since they don't even work with 100s demonstrations in simulation.

---

### Official Review · Reviewer_hXJH · 2022-07-30

**Originality:** Good
**Technical Quality:** Good
**Clarity Of Presentation:** Very Good
**Impact:** 3

**Recommendation:**

Weak Accept: I recommend accepting the paper, but will not argue for my recommendation if the majority of other reviewers have a different opinion.

**Summary:**

This paper proposes a language-guided robot manipulation framework built on top of Perceiver IO. The proposed framework – Perceiver-Actor or PerAct in short, performs multi-stage manipulation tasks in a multi-task manner, showing superior performance in both simulated environments (RLBench) and in real-world settings trained with few demonstrations.

**Issues:**

See weaknesses.

**Quality Of The Limitations Section:**

Limitations are not well addressed

**Reviewer Expertise:**

4: The reviewer is confident but not absolutely certain that the evaluation is correct

**Robotics Focus:**

Sufficient demonstration on hardware

**Strengths And Weaknesses:**

Strength: This paper shows a promising direction to scale up vision-based manipulation tasks with transformers. It combines modern large-scale pre-training frameworks, using CLIP to encode language instructions, and using Perceiver IO for learning discretized actions. Using Perceiver IO is significantly helpful to reduce memory required for encoding high-resolution colored voxels inputs, (attended with a much smaller latent embedding) compared to previous methods like C2FARM alternatively require to multi-stage (coarse-to-fine) to encode fine-level voxels with limited memory.

Weakness: Though the proposed framework is cleaner in design, it poses some other limitations.

-- The training setting for keyframe discovery and learning Q-attention values is pretty much following ARM and C2F ARM. The authors did mention in keyframe discovery but missed the related references in action parametrization (L162 - 170).

-- The training time requirement is significantly larger than the related works. In L185-186, the authors stated that the training needs 8 V100 for 16 days (ImageNet experiments with the same amount of compute takes only 2 days, but with 1M training data, here the number of training data would be definitely less than 100k), whilst C2F ARM only requires less than 1 hour for each task in a single GPU. Considering each task is only augmented with 10 demonstrations, I hope the authors could slightly explain why training takes such long time? Is it simply because transformers in general take longer time to train?

-- Design details in baseline methods. Are two mentioned baseline methods: Image-BC / C2FARM-BC single-task or multi-task learning methods? L223-224 indicates all methods are multi-task agents, how the sense of “task” then is fed to these baselines? Or the authors just treat everything is an image-to-action type of learning tasks without introducing the idea of task in training? This is slightly unfair, since the sense of task is encoded in language in the proposed framework.

-- Compared to other multi-task methods. Multi-task robot learning has also been heavily investigated in the multi-task learning community. For example, notable works like: PCGrad [1], Auto-Lambda [2] also evaluated multi-task robot learning which might be more suitable baselines compared to Image-BC / C2F ARM-BC which were designed as single-task learning methods. In particular, Auto-Lambda also performs multi-task learning on RLBench (though not language guided), but with the performance of same evaluated tasks like: slide block: 77 (72), money in safe: 64 (44), place wine: 19 (12), push buttons: 95 (48), trained with the same number of 100 demonstrations, achieved better performance than the proposed method. I would suggest to do a thorough literature review on multi-task learning on manipulation and should compare with at least 1 or 2 related baselines designed specifically for multi-task manipulation.

[1] Yu et al.  Gradient Surgery for Multi-Task Learning. NeurIPS 2020

[2] Liu et al. Auto-Lambda: Disentangling Dynamic Task Relationships.  TMLR 2022.


**Summary Of Recommendation:**

In general, I quite like the idea as it shows a promising direction to bring transformers to the manipulation community. I would further raise the score if the authors could resolve my concerns stated in the weaknesses section.

---

> ### Author Response · Authors · 2022-08-22
> **Response to Reviewer-hXJH**
>
> We thank the reviewer for their feedback. We are glad the reviewer finds the approach promising. But there are a lot of misunderstandings in the Weaknesses section. We hope our responses help clarify these misunderstandings. Please reach out to us if you have any further questions.
>
> **Citing C2FARM for action parameterization**
>
> This is a good point. We will cite C2FARM in the action parameterization description.
>
> **Why is training PerAct so slow?**
>
> _“C2F ARM only requires less than 1 hour for each task in a single GPU”_ -> C2FARM takes 1 hr to train on 1 task variation. PerAct is a single-agent trained on 249 task variations (across 18 tasks). The longer training time is predominantly due to there being 249 times more tasks in the dataset, and also the very difficult problem of jointly optimizing on 249 variations. Infact, our C2FARM baseline in Table 2 was also trained for 11 days with several GPUs. Training PerAct’s Transformer is slightly slower than C2FARM, but we believe the benefits (performance and receptive field) vastly outweigh the cost of offline training for a few extra days. Further, in single-variation settings, it’s possible to train PerAct within 1.5-2hrs, just like C2FARM. We acknowledge that the paper could have done a better job of being upfront on the size of task variations. We will update the paper to reflect this.
>
> _“ImageNet experiments with the same amount of compute takes only 2 days"_-> The comparison to ImageNet here is a bit unfair. ImageNet has been active since 2009, and several efforts have amortized the time for data loading, preprocessing, and batching with nearly a decade's worth of engineering. PerAct’s robot learning framework is still in its infancy, with data loading and voxelization taking up most of the overhead. Of the 249 task variations, each demonstration contains 100s of individual frames, and each frame contains 4 RGB-D images, all of which have to be aggregated into a grid of 1 million voxels for each datapoint. The training pipeline here is vastly different from loading small 2D images, for which there is a lot of well-established infrastructure. During this work, we did not spend a significant amount of time optimizing the training pipeline, and frankly the authors do not have the engineering expertise required, but we are optimistic that training time can be dramatically reduced in the future with some effort. We also note that the biggest Transformers in NLP like GPT3 and BLOOM take months to train.
>
> **How is the task specified for baseline methods?**
>
> All baselines were given language goals like with PerAct (except the PerAct w/o lang ablation). As stated in L227, we use FiLM-based language conditioning for Image-BC following BC-Z. And as stated in L235, we use LingUNets for C2FARM where we condition the bottleneck features of C2FARM’s UNet with language encodings. We acknowledge that some of these details were not clear in the submission. We will update the paper accordingly.
>
> **Comparing against orthogonal multi-task optimization methods**
>
> Thank you for pointing us to PCGrad and Auto-Lambda. The authors are aware of these works, but **we strongly believe that these optimization methods are orthogonal to our approach and goals**. PerAct’s key goal is “training Transformers in the real-world for BC with limited data and high-dimensional input”, and NOT “what’s the best way of doing multi-task optimization”.
>
> It is true that Auto-Lambda achieves better performance with ARM on RLBench by proposing a multi-task optimization framework that goes beyond just equal-weight task sampling. But Auto-Lambda and PCGrad _could be easily applied to PerAct as well_ to improve multi-task optimization. So we are not sure what is the purpose of evaluating against Auto-Lambda and PC-Grad, which just adds a **tangential dimension to the evaluation results**. Please note that it’s also practically infeasible to add tangential dimensions to large-scale experiments that already take several weeks to train and evaluate.
>
> Further, the quantitative numbers from AutoLambda, i.e 77 (72) slide_block, 19 (12) place_wine etc. are not comparable to PerAct’s results. AutoLambda was trained on 10 task variations, PerAct was trained on 249 variations. slide_block in PerAct was trained and evaluated on 4 task variations instead of just 1 in AutoLambda. We kindly invite the reviewer to explore the simulation results on our interactive website: [peract.github.io](https://peract.github.io/) to get a sense of the additional task variations. We also humbly request the reviewer to check out the appendix for details on how RLBench tasks were modified to introduce more complexity in tasks. We believe PerAct was trained on significantly more RLBench tasks and task variations than any other prior multi-task work (that we are aware of). We will also add a discussion on AutoLamba and other concurrent works to explicitly clarify this.

---

> > ### Comment · Reviewer_hXJH · 2022-08-23
> > **Response to the rebuttal**
> >
> > We'd like to thank the authors for their detailed rebuttal.
> >
> > Here are the follow-up comments.
> >
> > **Why is training PerAct so slow?**
> >
> > Thanks for the explanation. Now it makes sense. I think it's better to give some explicit definitions of tasks and variations at the beginning of the method section, as other reviewers were also confused about the training data. So I assume the total number of data required in your training is 18 tasks * 6 - 90 variations which give 249 variations * 100 demonstrations, am I right? So say you were not using the coarse-to-fine training strategy, processing these voxels would be quite computationally expensive.
> >
> > But I don't think it's a good argument to bring in GPT3 and BLOOM. They are in another level of scale of data and network size.
> >
> > **How is the task specified for baseline methods?**
> >
> > Thanks for the clarification. I would suggest to add one short paragraph to clearly distinguish the language embedding/conditioning method used in Preact and the ones used in Image-BC and C2FARM-BC. Are all language embeddings coming from CLIP? These details are essential but missing in the main text and I hope the authors could update the new paper with these training details.
> >
> > **Comparing against orthogonal multi-task optimization methods**
> >
> > I admit that I misread this part of the contribution in my first paper evaluation. And I agree with the authors that comparing multi-task methods is not necessary. My misunderstanding was mainly from that I thought the baseline methods were not language-based, but task-embedding-based multi-task methods. As the method name in Table 2 is the same as the original work, and there are not a lot of places emphasizing these are "language-conditioned" multi-task methods, it's easy to miss the details from the first reading. But all my confusions are clear.
> >
> > But I would agree emphasizing that the higher complexity of task variations compared to Auto-Lambda and slightly discussing the relationships with other multi-task robot learning frameworks will be helpful and definitely make this paper stronger.
> >
> > A small note: In your website Results Section, are the "instances" here is the same meaning of "variations" in the paper? If so, please keep a consistent notations.

---

> > > ### Author Response · Authors · 2022-08-24
> > > **Thanks for the follow-up!**
> > >
> > > Thank you so much for the quick follow-up! We are glad most of the major confusions have been cleared up.
> > >
> > > _“as other reviewers were also confused about the training data”_
> > >
> > > We completely understand. The original submission did a poor job of communicating the size of the training dataset.
> > >
> > > _“So I assume the total number of data required in your training is 18 tasks * 6 - 90 variations which give 249 variations * 100 demonstrations, am I right?”_
> > >
> > > Yes, but each demonstration also contains 100s of individual frames since it’s recorded at ~30Hz. Since we follow the “predict the next best keyframe” method from James et al. (see Appendix F), each demonstration could potentially have 100s of individual training data points for supervising the classifier.
> > >
> > > _“So say you were not using the coarse-to-fine training strategy, processing these voxels would be quite computationally expensive.”_
> > >
> > > Yes. It’s true that the coarse-to-fine strategy would reduce the training time by reducing the input dimensions. But for a lot of tasks, the coarse-to-fine has such a restrictive receptive field that they might not be solvable without global information. See [this example](https://peract.github.io/media/results/qpred/stick.mp4) where the agent has to “hit a green ball with the stick”. After grasping the long stick (at 0:39 in the video), the predicted actions are more than half-way across the scene to where the stick is actually interacting with the ball. As such, the finest-level of a C2FARM agent won’t be able to see the entire stick, and at best would have to memorize locations/directions to move the stick without knowing where the “green ball” is at.
> > >
> > > _“I would suggest to add one short paragraph to clearly distinguish the language embedding/conditioning method”_
> > >
> > > Absolutely. We are working on an update with additional baseline details.
> > >
> > > _“Are all language embeddings coming from CLIP?”_
> > >
> > > Yes. This is a good technical detail to highlight. Thanks!
> > >
> > > _“But I would agree emphasizing that the higher complexity of task variations compared to Auto-Lambda and slightly discussing the relationships with other multi-task robot learning frameworks will be helpful and definitely make this paper stronger”_
> > >
> > > Agreed!
> > >
> > > _“As the method name in Table 2 is the same as the original work,”_
> > >
> > > Good point.
> > >
> > > _“A small note: In your website Results Section, are the "instances" here is the same meaning of "variations" in the paper? If so, please keep a consistent notations.”_
> > >
> > > Good catch! We will fix this. Thanks.
> > >
> > > Once again we thank you for your valuable feedback. Your suggestions have helped us improve the paper.

---

> > > > ### Comment · Reviewer_hXJH · 2022-08-24
> > > > **Final Question**
> > > >
> > > > I am still a bit confused on the training data.
> > > >
> > > > If you strictly followed the data pre-processing strategy in ARM / C2FARM, then the only data you need to supervise the classifier is the detected keyframes which should be very sparse per demonstration. As you also mentioned in the paper, each demonstration will produce only 2 - 17 frames. So why you need to record in all frames in 30 Hz? I thought the intermediate trajectories will be automatically solved using the RLBench embedded planner. At least this is what ARM did based on my expereience.
> > > >
> > > > Also, though I remember I did see the appendix in the paper, but somehow I cannot access anymore. The current pdf only contains 13 pages up to the reference section.

---

> > > > > ### Author Response · Authors · 2022-08-25
> > > > > **Demo Augmentation**
> > > > >
> > > > > No worries! Here is the full paper with appendix: [https://peract.github.io/paper/peract_corl2022.pdf](https://peract.github.io/paper/peract_corl2022.pdf)
> > > > >
> > > > > [Figure 9](https://i.ibb.co/54bTwHB/Screen-Shot-2022-08-24-at-5-17-38-PM.png) in the appendix might be helpful here. In this example, there are only two keyframes – k1 and k2. But every datapoint in the recorded trajectory – the orange points – can be used to supervise the action detection task. Each datapoint will have the next keyframe action as the expected action given the corresponding voxelized observation.
> > > > >
> > > > > This demo augmentation method allows us to easily get lots of supervision data from a single demonstration/trajectory. And this allows us to train robust action-detectors like in this [tracking demo](https://peract.github.io/media/results/animations/handsan_tracking_v2.mp4), which works despite noisy voxel input from just 5 training demos.

---

> > > > > > ### Comment · Reviewer_hXJH · 2022-08-25
> > > > > > **Response**
> > > > > >
> > > > > > Thanks for the explanation. All my concerns have been resolved.

---

### Official Review · Reviewer_xg2V · 2022-07-30

**Originality:** Very Good
**Technical Quality:** Very Good
**Clarity Of Presentation:** Good
**Impact:** 4

**Recommendation:**

Strong Accept: I recommend accepting the paper and will argue for my recommendation even if other reviewers hold a different opinion.

**Summary:**

The paper introduces a language-conditioned behavior-cloning agent for multi-task 6-DoF visual manipulation. In Leyman terms, the paper is a variation of the Coarse-to-fine ARM paper, where the key differences are:

- (1) The addition of the Perceiver Transformer [1], rather an a 3D convolution.
- (2) Using behavior cloning rather than reinforcement learning.
- (3) Language conditioning to enable multi task/variation learning.  The paper evaluated in both simulation (on 18 RLBench task) and in 7 real world tasks.

**Issues:**

See weaknesses

**Quality Of The Limitations Section:**

Limitations are addressed clearly

**Reviewer Expertise:**

5: The reviewer is absolutely certain that the evaluation is correct and very familiar with the relevant literature

**Robotics Focus:**

Sufficient demonstration on hardware

**Strengths And Weaknesses:**

**Strengths**

- An efficient BC algorithm that performs well on 53 demonstrations.
- A large number of simulation experiments for reproducibility, and also real world experiments to show that the method is suitable for real-word training.
- The authors did well to state clearly any modifications to the tasks in RLBench , ensuring reproducibility.
- A detailed description of the limitations (in appendix).
- The addition of the perceiver seems to clearly help learning these complex tasks.
- Fig 2 gives a good overview of the system.
- Other than the introduction, which could be a little clearer, the rest of the paper is well written.

**Weaknesses**

- The introduction could be improved to better frame the proposed work. There seems to be quite a lot of similarities with the C2F-ARM work through the paper (next-best pose action, 3D voxelization, motion planning, demo aug, keyframe discovery) but it would be good to perhaps front-load a summary of these similarities and differences, rather than having the reader have to go through the entire paper. Try to make it instantly clear what separates this work.

- There isn’t anything in the paper that is BC specific that has been introduced in this paper, and so it would have been nice to include results of the effect of adding the transformer for the RL agent, rather than just the BC one.

- Ablation on patch size (other than the 5^3 used) would be very useful.


**Other minor comments**

- ln 69: “… has a limited receptive field that cannot look at the entire scene at the finest level”. This is not exactly true; worth noting that this can be eleviated with [b].

- ln 71:  “PERACT does BC instead of RL, which enables us to easily train a multi-task agent for several tasks”, but [a] has trained a multi-task policy with C2F-ARM. I’d suggest amending this comment and elaborate on how these two works differ. For one, the PERACT can do both multi task and multi-variation because it conditions on language, while [a] only does multi task because it does not use task conditioning.

- [c] is relevant work on using a next-best action space for multi-task BC on RLBench tasks.

- Fig 2: Could benefit from having the RGB views to make it clear what is in the voxel grid scene.

- ln 136: [71] should be cited when discussing collide prediction.

- I wonder if PERACT training/memory overhead could be reduced by applying a coarse-to-fine structure as in C2F-ARM? Fig 4 evaluated multiple hierarchies for C2F-ARM, and it would be nice to see something similar also for PERACT. If we can get similar performance from having this hierarchy, but reduce the memory required, then that would be great.

[a] Mandi et al.  "On the Effectiveness of Fine-tuning Versus Meta-reinforcement Learning."

[b] James at al. "Coarse-to-fine Q-attention with Tree Expansion."

[c] Liu et al. "Auto-Lambda: Disentangling Dynamic Task Relationships."

**Summary Of Recommendation:**

I believe the work would be valuable to the community. I would consider increasing my recommendation if the majority of the issues are addressed in the rebuttal phase.

---

> ### Author Response · Authors · 2022-08-22
> **Response to Reviewer-xg2V - Part 1**
>
> We thank the reviewer for their insightful comments. We are glad the reviewer appreciates the efficiency of the approach and the thoroughness of the technical presentation.
>
> **Better positioning with respect to C2FARM**
>
> This is a good point, and we agree with the reviewer that the introduction could do a better job positioning PerAct with respect to C2FARM. We will update the introduction to make this clearer.
>
> **Results with RL**
>
> For the original paper submission, we were also interested in training PerAct with RL. However, there were a number of practical challenges. (1) It takes 16 days to train an offline BC agent, so an RL agent that has to interact with the simulator is significantly slower. Especially since we are training a single agent on 249 task variations, and some of these tasks involve long time-horizons like 15 keyframe actions. (2) In the spirit of benchmarking against C2FARM (which stores demos and online interactions in one large replay buffer), we found that multi-task training consumes significantly more memory to store transitions, especially considering our large observations. That being said, we are working on mitigating these issues, and we might release some RL baselines in our code-release, if possible. At least in single-task and single-variation settings, PerAct+RL might be feasible. We are glad the reviewer appreciates that the contributions of PerAct go beyond BC, and we thank the reviewer for highlighting this.
>
> **Ablation on patch-sizes**
>
> This is a good ablation to investigate. We just started training agents with different patch-sizes. We will update the appendix with new results whenever they are ready (which will likely be after the rebuttal deadline).
>
> _ln 69: “… has a limited receptive field that cannot look at the entire scene at the finest level”. This is not exactly true; worth noting that this can be eleviated with [b]_
>
> Thanks for the pointer to [b]. But the tree-expansion in [b] does not elevate the receptive field problem _at the finest-level_. Please see [this example](https://peract.github.io/media/results/qpred/stick.mp4) where the agent has to “hit a green ball with the stick”. After grasping the long stick (at 0:39 in the video), the predicted actions are more than half-way across the scene to where the stick is actually interacting with the ball. As such, even with [b], the finest-level of a C2FARM agent won’t be able to see the entire stick, and at best would have to memorize locations/directions to move the stick.
>
> _ln 71: “PERACT does BC instead of RL, which enables us to easily train a multi-task agent for several tasks”, but [a] has trained a multi-task policy with C2F-ARM. I’d suggest amending this comment and elaborate on how these two works differ. For one, the PERACT can do both multi task and multi-variation because it conditions on language, while [a] only does multi task because it does not use task conditioning._
>
> Thanks for the pointer to [a]. Please note that [a] is unpublished work that was **arXived 8 days before the CoRL submission deadline**. We understand that this is a very competitive space, so will add a “Concurrent Works'' section in the appendix.
>
> _[c] is relevant work on using a next-best action space for multi-task BC on RLBench tasks._
>
> Thanks for the pointer. We will cite [c] for next-best action space and also discuss it in the limitations section on better approaches for multi-task optimization.
>
> _Fig 2: Could benefit from having the RGB views to make it clear what is in the voxel grid scene._
>
> This is a good suggestion. We will update the figure.
>
> _ln 136: [71] should be cited when discussing collide prediction._
>
> Please note that we do not use learned path-ranking from [71] for collision avoidance or motion-planning. As stated in Section 3.2, we simply predict a binary variable that indicates if the motion planner is allowed to collide with objects or not when coming up with paths. In the real-world, this means either providing the full occupancy grid (from the voxelization) to the RRT-based motion planner or an empty occupancy grid. We do cite [71] in the limitations section when discussing future works that could potentially learn paths instead of always relying on a sampling-based motion planner.

---

> > ### Author Response · Authors · 2022-08-22
> > **Response to Reviewer-xg2V - Part 2**
> >
> > _I wonder if PERACT training/memory overhead could be reduced by applying a coarse-to-fine structure as in C2F-ARM? Fig 4 evaluated multiple hierarchies for C2F-ARM, and it would be nice to see something similar also for PERACT. If we can get similar performance from having this hierarchy, but reduce the memory required, then that would be great._
> >
> > This might be possible. Recent works like Hierarchical Perceiver ([https://arxiv.org/abs/2202.10890](https://arxiv.org/abs/2202.10890)) could be relevant here. But in PerAct, we wanted to keep the architecture as simple as possible. With vision transformers, simple architectures like ViT have been remarkably robust and prevalent despite advancements in memory and training efficiency. PerAct is essentially the 3D equivalent of ViT.

---

> > > ### Comment · Reviewer_xg2V · 2022-08-26
> > > **Response to rebuttal**
> > >
> > > I thank the authors for addressing questions and comments. I've also read the revision, and the paper has been improved.
> > >
> > > I like the paper, and I think it will be a great addition to CoRL. I'll raise my rating from weak accept to strong accept during phase 2.

---

### Official Review · Reviewer_vWwj · 2022-08-03

**Originality:** Very Good
**Technical Quality:** Good
**Clarity Of Presentation:** Very Good
**Impact:** 4

**Recommendation:**

Weak Accept: I recommend accepting the paper, but will not argue for my recommendation if the majority of other reviewers have a different opinion.

**Summary:**

This paper proposes to use a Transformer to perform multi-task manipulation via language-conditioned behavior-cloning. The key challenge is Transformer is known to be data inefficient. By casting behavior-cloning as an action detection task, PREACT achieves great efficiency on a range of simulated and real-world tasks.

**Issues:**

Strength:
* L45, please consider defining action-centric representation when it first shows up.
* L125, please consider providing an intuitive understanding of keyframes, e.g. "the pose of the end-effector at the point where object interaction should begin"
* L174, one task may require several keyframes to finish, are the language goals remain the same?


**Quality Of The Limitations Section:**

Additional details required

**Reviewer Expertise:**

4: The reviewer is confident but not absolutely certain that the evaluation is correct

**Robotics Focus:**

Sufficient demonstration on hardware

**Strengths And Weaknesses:**

Strength:
* The paper is well-organized and easy to follow.
* The proposed method is succinct and easy to understand.
* The tasks are challenging, and the proposed method achieves good performance while maintaining good data efficiency.
* The real robot experiments make the paper promising.

Suggestions and Improvements：
* As the author mentioned, both Transformers and 3D Convs are data inefficient, but I did not see why detecting voxel actions is extremely data efficient (i.e. achieving ~$60\%$ trained with only 53 demos across 7 tasks.) Could you provide more analysis?
* It is hard to evaluate the generalizability of PREACT. As mentioned in L213, all objects are seen during training. Could you please provide more detailed differences between training and testing? (e.g. the size differences as shown on the website; training: stack yellow block on blue block, testing: stack red block on yellow block; training: turn left tap, testing: turn right tap, etc)

**Summary Of Recommendation:**

I'd recommend "weak accept". The paper studies difficult problems and achieves good performance. It would be great if the author can provide more analysis about why action detection (classification) makes such a huge difference from regression (BC).

---

> ### Author Response · Authors · 2022-08-22
> **Response to Reviewer-vWwj**
>
> Thank you for the insightful feedback. We are glad the reviewer finds the approach promising and that the paper is easy to read.
>
> **Why is PerAct dramatically more efficient?**
>
> Compared to image-based baselines like Image-BC, PerAct has a strong 3D prior for learning 6-DoF policies. A naive image-based agent has to learn hand-eye coordination and multi-view aggregation, which are both data intensive. Instead, PerAct’s voxelized observation and action space naturally fits with the 3D nature of 6-DoF manipulation, especially since the observation and action spaces are aligned. With regards to 3D ConvNets (C2FARM), the performance gains are likely due to the larger receptive field of Transformers – we study this in Section 4.3. We thank the reviewer for bringing up this point, and we will better highlight it in the paper.
>
> **Why is classification more efficient than regression?**
>
> We do not have a strong theoretical understanding of this phenomenon. But in practice, discrete classification is often more efficient than continuous regression. It’s easier to get a good coverage of discrete spaces in the training data since discrete spaces are finite and limited. Whereas continuous spaces are unconstrained and much larger, often require extrapolation beyond the training distribution. Further, training classifiers with cross-entropy loss also allows us easily to represent multi-modal action distributions, where there could be more than one valid action. For example, in "place 3 mugs" from [this screenshot](https://i.ibb.co/Gcx8b01/Screen-Shot-2022-08-22-at-12-52-50-PM.png), picking any of the three mugs would be valid.
>
> **How generalizable is PerAct?**
>
> We agree with the reviewer that the paper could have done a better job of explaining the difference between train and test scenarios. As stated in L213, the focus of this work is to learn a single policy for 100s of task variants. Due to the design of RLBench, there are no evaluations on unseen objects or instances. As such, in the 100 demonstrations setting, the agent has seen all the objects (and their associated nouns), but has to generalize to unseen poses and randomly sampled goals from randomly sampled tasks. In the 10 demonstration setting, it’s likely that the agent has not seen certain objects attributes (like ‘teal color’) since the training demonstrations are 10 randomly sampled task variants, and some tasks have more than 10 variations. We will update the paper to explain this better.
>
> More broadly, we note that PerAct is essentially a classifier trained to ‘detect actions’. We hypothesize that it might be feasible to train PerAct on a broad distribution of objects (like in MaskRCNN, YOLO etc.) and generalize to unseen instances, much like standard object detectors in computer vision. For this paper, we found it practically difficult and time consuming to create/add new assets to RLBench, so we leave this investigation for future work.
>
> **_L45, please consider defining action-centric representation when it first shows up._**
>
> Good point. Thank you!
>
> **_L125, please consider providing an intuitive understanding of keyframes, e.g. "the pose of the end-effector at the point where object interaction should begin"_**
>
> Thanks for the pointer. We will add this.
>
> **_L174, one task may require several keyframes to finish, are the language goals remain the same?_**
>
> Yes, the language goals are the same throughout all keyframes. We will make this clear.

---

### Author Response · Authors · 2022-08-22
**General response to all reviewers**

We thank all reviewers for their feedback. We are glad everyone appreciates PerAct’s approach, strong empirical results, and the technical presentation of the work. Here we highlight some common themes across reviews.

**Better positioning with respect to prior works**

We agree with all reviewers that the paper could have done a better job of positioning PerAct with respect to C2FARM. We will update the paper to fix this.

**Exploring orthogonal research directions**

We are excited that most reviewers find PerAct’s approach exciting and rated it as “… contains interesting new ideas that will potentially have a major impact in robotics or machine learning”. In fact, most reviewers proposed a unique and independent orthogonal direction for future work:


* Reviewer-xg2V is interested in training in PerAct with RL, since our method is not just specific to BC.
* Reviewer-hXJH is interested in improving multi-task optimization by going beyond uniform task-sampling.
* Reviewer-YHWS mentioned language-based planning approaches like SayCan for composing manipulation skills for sequential tasks.

All these orthogonal directions are very exciting and ripe for future work. But for this paper, we reiterate that our focus is to “get a multi-task Transformer to work for BC with limited data and high-dimensional input”. We hope our large-scale experiments in sim, and validation experiments in the real-world, sufficiently support this premise.

**Scale of RLBench experiments**

One common theme we noticed is that the paper did a poor job of communicating how many RLBench task variations PerAct was trained on. The training and evaluation dataset included a total of 249 task variations. We believe this is substantially larger than in any prior work (that we are aware of). We invite reviewers to explore the simulation results with the interactive explorer on our website: [https://peract.github.io/](https://peract.github.io/) to get a sense of the number of task variations and the complexity involved with each variation. We will also update the paper to better communicate this.

**“Limitations are not well addressed”**

Reviewer-vWwj and Reviewer-hXJH requested additional details on the limitations of PerAct. Please note that the original submission contains ~2 pages of limitations and “things that did not work” at the bottom of the appendix. We hope this is sufficient. We tried discussing more limitations in the main paper, but squeezing ~2 pages of limitations within 8 pages is quite challenging.

---

### Author Response · Authors · 2022-08-25
**Paper Update**

We again thank all the reviewers for their insightful comments. All the feedback has helped us greatly improve the paper.

Updates for the rebuttal:



* All the changes/updates promised in the responses below. The changes and relevant sections are **highlighted in blue**.
* A dedicated appendix section (Appendix I) for additional related work from the discussions below. These include: multi-task learning, concurrent works, and language-based planning literature, all with a brief discussion.
* Patch-size ablations are still running.

General updates to improve the paper:



* New RLBench results with image-to-action agents that use a ViT backbone
* New perturbation tests in Appendix L that study generalization to object instances
* New results on training single-task agents for high-precision tasks in Appendix H
* New write-up on “Emergent Properties” of PerAct in Appendix M
* Real-time [tracking demo](https://peract.github.io/media/results/animations/handsan_tracking_v2.mp4) with PerAct.
* Updated website with interactive results explorer: [https://peract.github.io/](https://peract.github.io/)

Please reach out to us if you have any more questions.

---

### Author Response · Authors · 2022-11-11
**Camera Ready - Quick Updates**

We again thank the reviewers and meta reviewer for their valuable feedback!

Some quick camera-ready updates:
- Added more related work.
- Added more sections in Limitations.
- Fixed error in average improvement calculation.
- Colab Tutorial on implementing and training PerAct from scratch is available: https://peract.github.io/
- Our modifications to RLBench are available: https://github.com/MohitShridhar/RLBench/tree/peract
- We are still working on the full code-release. Currently, we are trying to make multi-GPU training easier and cleaner to implement.

---

> ### Author Response · Authors · 2022-11-25
> **Code release**
>
> Apologies for the delay. The full code, pre-generated datasets, and pre-trained checkpoints are now available at: https://github.com/peract/peract
>
> We look forward to exciting future works in this area!

---

### Meta-Review · Area_Chair_rrTt · 2022-08-02

**Recommendation:** Accept (Poster)
**Confidence:** 4

**Metareview:**

Strengths:
- The contributions are interesting and relevant to the CoRL community
- The experiments are generally convincing.
- The paper includes experiments on a real robot
- The paper is well-written
- There is a detailed discussion of the limitations in the appendix

Weaknesses:
- There are a lot of similarities with the ARM and C2F-ARM works, which is okay -- much of research is incremental on top of prior work. But, the paper could have done a better job at framing the contributions in the context of these prior works.
- There are some additions to the experiments that would make them more thorough and informative, e.g. including comparisons on the real robot, including RL experiments, including ablations on patch size, and adding more comparisons in simulation such as to multi-task methods or SayCan.

The author response and discussion has helped address the concerns. The paper makes a valuable contribution to the CoRL conference.

**Best Paper Nomination:**

No

---

> ### Author Response · Authors · 2022-08-22
> **Response to Meta Reviewer**
>
> We thank the meta-reviewer for a great summary of reviews and insightful comments.
>
> **Better positioning with respect to C2FARM and ARM**
>
> We agree that the paper could have done a better job of positioning the contributions with respect to C2FARM and ARM. We will update the introduction and relevant sections to reflect this.
>
> **Regarding ‘incremental’ work**
>
> In PerAct, we train a Transformer _from scratch_ on several tasks with an input of 1 million voxels. And it works in the real-world. We are not aware of any prior works that got Transformers to work robustly for manipulation with such limited data, on such high-dimensional inputs, and on such diverse tasks. Additionally, in RLBench, PerAct outperforms C2FARM (a SOTA ConvNet method) on 18 tasks **by 1.68x (+168%) with 10 demos and 2.66x (+266%) with 100 demos **as stated in L244**.** We hope 2.66x improvement over C2FARM across 18 tasks counts for something more than just “incremental”. We acknowledge that the paper could have done a better job of highlighting these capabilities and performance gains, and we will update the paper to fix this.
>
> **Additional Experiments**
>
> _Different Patch-sizes_
>
> We just started experiments with different patch-sizes. We will add the new results to the paper as soon as they are ready.
>
>
> _RL Baselines_
>
> There are some technical difficulties with training a single RL agent on 249 task variations. Online interactions in the V-REP simulator (the simulator underlying RLBench) are very slow, and some tasks have very long time-horizons. Please see the response to Reviewer-xg2V for additional details. We will try to mitigate these issues, and maybe release some RL baselines with the code-release, if possible.
>
>
> _Comparisons with real-robot experiments_
>
> One-to-one comparisons across frameworks in the real-world is very difficult and often infeasible. It’s tedious (and sometimes impossible) to reproduce the exact same initial conditions with objects, object poses, lighting, and scene configuration across evaluations. We believe the simulation results sufficiently compare against prior works in a fair and reproducible manner. The main purpose of our real-world experiments is to demonstrate that PerAct _does_ actually work in the real-world. Please see the response to Reviewer-YHWS for additional details.
>
>
> _Comparisons to multi-task optimization methods_
>
> The multi-task optimization methods highlighted by Reviewer-hXJH are **orthogonal** to the premise and goals of PerAct. Future works extending PerAct could go beyond uniformly sampling tasks for better multi-task optimization, but this is currently out-of-scope for this paper. This paper focuses on “getting Transformers to work for BC with limited data and high-dimensional input” and not “what’s the best way of doing multi-task optimization”. Please see the response to Reviewer-hXJH for details.
>
>
> _Comparisons to SayCan_
>
> SayCan combines pre-trained manipulation skills with a language-model for high-level planning. Language-based planning is **orthogonal** to PerAct’s approach. Future works could combine SayCan’s language-model (PaLM) with PerAct to execute a sequence of manipulation skills. But this is currently out-of-scope for this paper, which does not focus on “language-based planning”. Also, the manipulation policies in SayCan are based on BC-Z, which we study in Table 2 with the Image-BC baseline. Further, SayCan’s complete infrastructure is not reproducible. We do not have access to SayCan’s robot dataset, robot hardware, scenes, objects, Google engineers, financial resources, or the time to collect 68000 demonstrations over the span of several months. Please see the response to Reviewer-YHWS for additional details. We acknowledge that citing SayCan in the introduction might have caused some confusion. We will fix this.